# Dermatologist-like explainable AI enhances melanoma diagnosis accuracy: eye-tracking study

Tirtha Chanda [1], Sarah Haggenmueller[1], Tabea-Clara Bucher [1], Tim Holland-Letz[2], Harald Kittler [3], Philipp Tschandl [3], Markus V. Heppt [4], Carola Berking [4], Jochen S. Utikal[5,6,7], Bastian Schilling[8], Claudia Buerger[8], Cristian Navarrete-Dechent[9], Matthias Goebeler [10], Jakob Nikolas Kather [11,12], Carolin V. Schneider[13], Benjamin Durani[14], Hendrike Durani[14], Martin Jansen[15], Juliane Wacker[16], Joerg Wacker[16], Reader Study Consortium* & Titus J. Brinker [1] ✉

Artificial intelligence (AI) systems substantially improve dermatologists' diagnostic accuracy for melanoma, with explainable AI (XAI) systems further enhancing their confidence and trust in AI-driven decisions. Despite these advancements, there remains a critical need for objective evaluation of how dermatologists engage with both AI and XAI tools. In this study, 76 dermatologists participate in a reader study, diagnosing 16 dermoscopic images of melanomas and nevi using an XAI system that provides detailed, domain-specific explanations, while eye-tracking technology assesses their interactions. Diagnostic performance is compared with that of a standard AI system lacking explanatory features. Here we show that XAI significantly improves dermatologists' diagnostic balanced accuracy by 2.8 percentage points compared to standard AI. Moreover, diagnostic disagreements with AI/XAI systems and complex lesions are associated with elevated cognitive load, as evidenced by increased ocular fixations. These insights have significant implications for the design of AI/XAI tools for visual tasks in dermatology and the broader development of XAI in medical diagnostics.

Melanoma accounts for the majority of deaths attributed to skin cancer worldwide, with early detection and excision being crucial for a favorable prognosis[1]. Explainable Artificial Intelligence (XAI) is a growing field that has the potential to revolutionize the way dermatologists diagnose and treat skin conditions. XAI is an extension of artificial intelligence (AI) that focuses on developing algorithms and models that can provide transparent and/or interpretable explanations for their decisions and predictions[2–4]. The two primary branches of XAI techniques are (1) post-hoc algorithms that are designed to retrospectively explain the decisions from a given model, such as Grad-CAM[3] and others[5,6], and (2) inherently interpretable algorithms that are

designed to be intrinsically understandable, such as logistic regression[7] and others[8,9]. A diagnosis assistance system requires intuitive explanations tailored to dermatologists as they need to assess the quality of the machine suggestions for each image they diagnose[10,11]. A few recent dermatological XAI systems aim to close the interpretability gap through the use of concept-bottleneck models[12]. Such models are trained to predict the concepts that are used to distinguish between melanomas and nevi, such as the well-established Derm7pt[13]. Lucieri et al. used the expert annotated concepts from the PH2[14] and derm7pt[15] datasets to create an XAI that provides lesion-level explanations based on concept vectors[16]. Jalaboi et al. employed a

A full list of affiliations appears at the end of the paper. *A list of authors and their affiliations appears at the end of the paper. ✉e-mail: titus.brinker@dkfz.de

convolutional neural network architecture that was designed to include localisations into training on clinical images of skin lesions[17]. Additionally, they composed an ontology of clinically established terms to explain why the annotated regions are diagnostically relevant. Chanda et al. extended on these works and introduced a concept-bottleneck XAI that was trained to detect established characteristics to distinguish between melanomas and nevi[2].

In dermatology, XAI is used for skin cancer detection, where it can highlight skin regions relevant to the diagnosis and/or provide textual justifications for the prediction. XAI has the potential to improve the accuracy and reliability of the diagnostic process in the healthcare domain by improving user trust and acceptance[18–21]. Previous studies with XAI in dermatology have shown that dermatologists' diagnostic confidence and trust in AI systems increase when using XAI compared to traditional AI systems[2,22]. However, these findings were primarily based on subjective measures, such as self-reported confidence levels and trust ratings, which can be influenced by various factors, including individual biases and the desire to conform to perceived expectations[23,24].

To provide a more objective understanding of the impact of XAI in dermatology, it is essential to investigate how dermatologists interact with AI and XAI systems during their diagnostic process, particularly in terms of their attention to the provided explanations, and whether this attention correlates with diagnostic accuracy. Eye tracking has been shown to serve as a valuable tool in visual search patterns and assessing cognitive load, which refers to the mental effort required to process information and perform tasks[25,26]. The analysis of ocular parameters, such as the number of fixations and fixation durations, can provide insights into the cognitive demands placed on individuals while performing tasks. For instance, in the context of dermatology, fixation-based metrics offer indications of the level of interest or confusion experienced by participants when evaluating pigmented lesions[27]. Higher numbers of fixations may suggest uncertainty or difficulty in locating specific features, while longer fixation durations could imply challenges in comprehending the content or identifying relevant information[28].

Dreiseitl et al. used eye-tracking technology to record and analyze how dermatologists of different experience levels examined and diagnosed digital images of pigmented skin lesions[29]. The study involved 16 participants who were classified into three groups based on their dermoscopy training. The eye-tracking system recorded the gaze track and fixations of the participants while they examined 28 images. Experts were faster, more accurate, and more consistent in their diagnosis than novices and intermediates. They also spent less time and had fewer fixations on the images, indicating a more pattern-oriented approach. The authors suggested that eye-tracking analysis can be used to identify important diagnostic features and to optimize training for less experienced dermatologists. This study, however, did not involve an AI system. Kimeswenger et al. compared AI and board-certified pathologists in analyzing histological whole-slide images (WSI) using eye tracking[30] and found significant differences in how the AI and pathologists identified tumors, suggesting they prioritize different areas or features within the WSIs. It may indicate that the AI is capable of detecting subtle features potentially overlooked by human observers or that the AI is relying on features that are not intuitively interpretable to humans. Their work, however, analyzed pathologists and AI independently, rather than in tandem. Eye tracking technology can precisely capture where and for how long dermatologists focus their visual attention while using XAI systems. This facilitates a more comprehensive exploration of the cognitive processes involved in dermatologists' interactions with XAI, providing insight into their decision-making mechanisms and the impact of XAI on their diagnostic process. While the potential of XAI in dermatology is promising, it remains uncertain to what extent dermatologists use or ignore these technologies in their diagnostic process.

In this work, we address a research gap by employing eye-tracking technology to examine how dermatologists of varying experience levels interact with AI and XAI systems when diagnosing dermoscopic images. By gaining insights into the visual patterns and diagnostic strategies employed by dermatologists when utilizing AI for dermoscopy, we aim to enhance the understanding of the potential benefits and challenges associated with integrating AI into dermatological practice. To this end, we conduct a two-phase reader study (Fig. 1a) with 50 dermatologists to quantify the influence of classifier decisions in terms of dermatologists' diagnostic accuracy and their attention towards the classifier explanations. In the AI phase, the dermatologists diagnose dermoscopic images of melanomas and nevi with AI support (Fig. 1b). In the XAI phase, they diagnose the images from the previous phase with XAI support (Fig. 1c). We leverage webcam-based eye-tracking to systematically analyze how dermatologists allocate their visual attention to XAI explanations and other components of the diagnostic process. To ensure the reliability and validity of our findings from the webcam-based eye-tracking experiments, we also conduct a validation study with an additional 25 dermatologists using a dedicated eye-tracking device, which offers greater precision than a webcam-based tracker. By comparing the two methods, we aim to establish the consistency of the dermatologists' visual attention patterns and to address any potential discrepancies between the two tracking systems. We show that XAI systems significantly improve balanced diagnostic accuracy compared to standard AI. Additionally, we find that diagnostic disagreements involving AI/XAI systems and complex lesions are linked to elevated cognitive load, as indicated by increased ocular fixations.

## Results

### Our XAI achieves good diagnostic accuracy

In our work, we aimed to investigate the interaction between dermatologists and an explainable artificial intelligence (XAI) system by analyzing how it impacted diagnostic accuracy and visual attention patterns. By leveraging both webcam-based and dedicated device-based eye-tracking, we aimed to uncover insights into the cognitive processes that dermatologists employ during skin cancer diagnosis, specifically focusing on how they interact with the explanations provided by XAI systems.

In our study, we required a classifier that not only made accurate predictions but also provided insights into the decision-making process. We extended the explainable classifier introduced in Chanda et al.[2] in our study. In their study, the authors introduced an XAI that provides domain-specific textual and region-based explanations for its predictions. To achieve this, they trained a classifier on explanations annotated by dermatologists. Therefore, the training set of their classifier comprised exclusively annotated images, and consequently its generalization performance on the diagnosis prediction between melanoma and nevus was limited. To address this, we introduced an additional output layer trained on both annotated and unannotated images, thereby improving the generalization performance. Details can be found in the Methods section.

Our XAI achieved a balanced accuracy of 86.5% (95% CI 83.2%, 90.0%) on the internal test set and 76.9% (95% CI 71.6%, 82.1%) on the external test set. In comparison, a baseline ResNet50 classifier achieved a balanced accuracy of 83.6% (95% CI 79.3%, 87.7%) on the internal test set and 77.1% (CI 95% 71.8%, 82.3%) on the external test set. Performance per characteristic is provided in Supplementary Fig. 1.

Thus, our XAI outperformed the baseline ResNet50 in internal test set accuracy and showed comparable performance on the external test set.

### Dermatologists' diagnostic accuracy increases with XAI over AI alone

We evaluated the impact of providing predictions alone (AI phase) versus providing explanations along with predictions (XAI phase) on

a

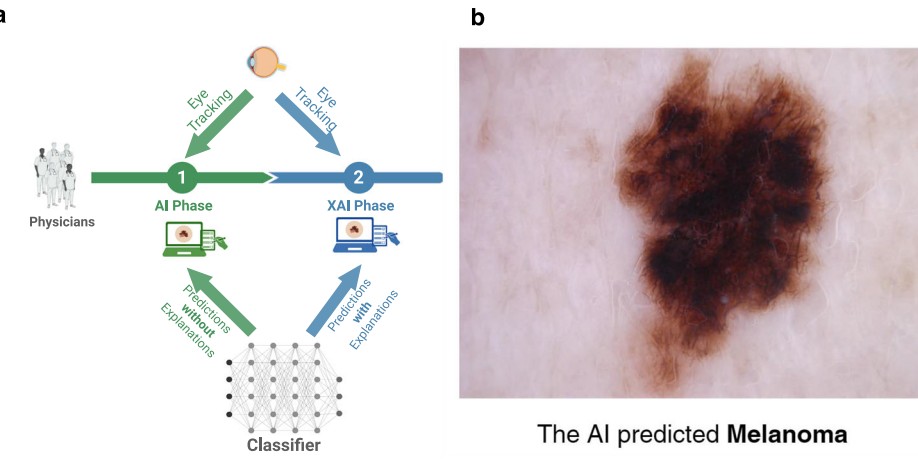

b

The AI predicted **Melanoma**

c

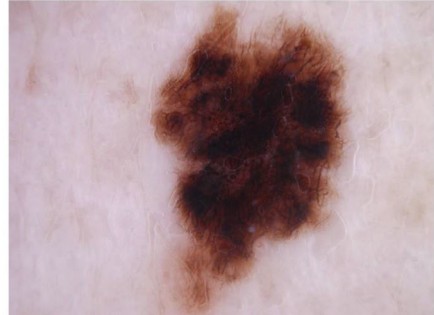
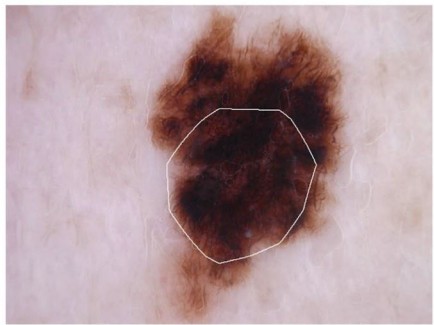

The AI predicted **Melanoma** because of *Pseudopods or radial lines at the lesion margin that do not occupy the entire lesional circumference*

**Fig. 1 | Schematic overview of the study design with AI and XAI prediction examples. a** Schematic overview of our two-phase reader study. Dermatologists were asked to diagnose 16 dermoscopic images each, consisting of melanomas and nevi. In the artificial intelligence (AI) phase, they were supported by an AI system that provided the predicted diagnoses for the images and were asked to provide their own diagnoses. In the explainable artificial intelligence (XAI) phase, they received support by an XAI that showed not only the predicted diagnoses but also the corresponding explanations. **b** An example dermoscopic image with the predicted diagnosis of the AI shown in the AI phase. **c** An example dermoscopic image, along with the predicted diagnosis from the XAI, and the corresponding textual and regional explanations provided during the XAI phase. Created in BioRender. Chanda, T. (2025).

the dermatologists' diagnostic accuracy. To further explore the benefits with XAI support over AI support, we conducted a correlation analysis between the extent of improvement in diagnostic accuracy and the dermatologists' self-reported level of expertise in dermoscopy.

Initially, we performed a combined analysis of both the webcam-based study and the device-based validation study. The results showed a mean dermatologist balanced accuracy (macro average of sensitivity and specificity) of 79.9% (95% CI 77.0–82.6%) with plain AI support and 82.7% (95% CI 80.3–85.0%) with XAI support (Fig. 2a, 2.8 percentage points improvement, $P = 0.013$, two-sided paired $t$-test, $n = 76$ dermatologists, Cohen's $d = 0.29$). Specifically, 34 dermatologists saw an improvement in performance, 20 experienced a decrease, and no change was observed for 22 dermatologists.

In the webcam-based study, the mean balanced accuracy was 77.8% (95% CI 74.3–81.3%) with AI support, increasing to 81.0% (95% CI 77.8–84.0%, 3.2% increase, $P = 0.018$, two-sided paired $t$-test, $n = 51$ dermatologists). In the device-based study, the mean balanced accuracy was slightly higher, at 84.0% (95% CI 80.0–88.0%) with AI support, and increased to 86.3% (95% CI 83.5–88.5%) with XAI support. However, the improvement was not significant ($P = 0.31$, two-sided paired $t$-test, $n = 25$ dermatologists).

We found no correlation between the dermatologists' experience levels and their increase in diagnostic accuracy with XAI over AI (Spearman's rank correlation −0.08, $P = 0.55$, $n = 61$ dermatologists)

(Fig. 2b). Details on dermatologist accuracies are provided in Supplementary Table 1.

Thus, providing XAI support resulted in a significant improvement in dermatologists' diagnostic accuracy compared to AI predictions alone, though this improvement was not correlated with their experience level.

## Disagreements with the classifier decisions correlate with ocular fixations

To determine the impact of dermatologist and classifier disagreements on the diagnostic process, we analyzed the number of fixations in cases where the dermatologist's diagnosis differed from the prediction of the classifier compared to cases where they aligned. Our findings indicate that in both AI and XAI phases, the mean fixation counts were higher when there was disagreement with the predictions of the classifier.

In the AI phase, the mean fixation count was 14.2 (95% CI 13.5–14.9) for cases where the classifier and the dermatologist agreed, and 19.6 (95% CI 17.8–21.4) for cases where they disagreed ($P < 0.001$, two-sided $t$-test, n_agreed = 644 cases, n_disagreed = 109 cases). Similarly, in the XAI phase, the mean fixation count was 16.7 (95% CI 15.9–17.5) for cases of agreement and 22.7 (95% CI 20.2–25.1) for cases of disagreement ($P < 0.001$, two-sided $t$-test, n_agreed = 658 cases, n_disagreed = 95 cases) (Fig. 3a). The mean fixation duration was 309.0 milliseconds (SD = 30.2 milliseconds).

**a**

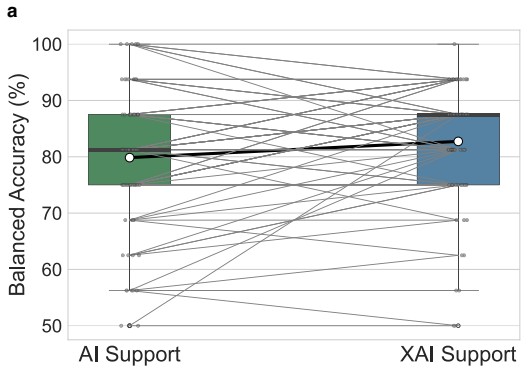

**b**

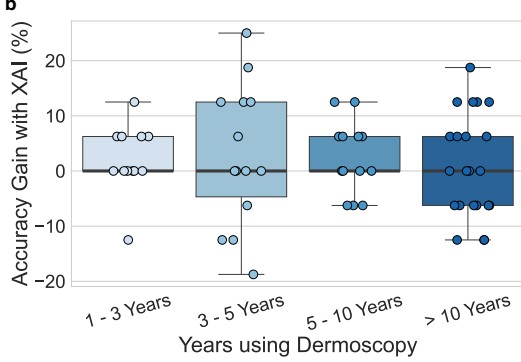

**Fig. 2 | Dermatologists' diagnostic accuracy with AI and XAI support.**
**a** Dermatologists' balanced accuracies with artificial intelligence (AI) support and explainable artificial intelligence (XAI) support ($P = 0.013$, two-sided paired $t$-test, $n = 76$ participants). The $y$-axis represents a continuous scale from 0 to 100 but is labeled at discrete intervals (e.g., 50, 60, etc.) for clarity. The gray lines between the boxes connect the same dermatologist between the AI and XAI phases, while the black lines indicate the means across all dermatologists. The horizontal line within each box denotes the median value, and the white dot represents the mean. The upper and lower box limits denote the 1st and 3rd quartiles, respectively, with the

whiskers extending to 1.5 times the interquartile range. **b** Numerical increase in dermatologists' diagnostic accuracy with XAI over AI (XAI phase accuracy minus AI phase accuracy) (two-sided Spearman's rank correlation $-0.08$, $P = 0.55$, $n = 61$ dermatologists). Each point represents one dermatologist. The horizontal line within each box denotes the median value, and the white dot represents the mean. The upper and lower box limits denote the 1st and 3rd quartiles, respectively, with the whiskers extending to 1.5 times the interquartile range. Source data are provided as a Source Data file.

To better understand these differences, we further analyzed the results for the webcam-based study and the device-based study. In the AI phase of the webcam-based study, the mean fixation count was 8.2 (95% CI 7.5–8.9) when there was agreement and 13.4 (95% CI 11.3–15.7) when there was disagreement ($P < 0.001$, two-sided $t$-test, n_agreed = 302 cases, n_disagreed = 51 cases). In the XAI phase, the mean fixation count increased to 9.8 (95% CI 8.8–10.8) for agreements and 17.1 (95% CI 13.9–20.6) for disagreements ($P < 0.001$, two-sided $t$-test, n_agreed = 303 cases, n_disagreed = 50 cases).

The device-based study showed similar trends. During the AI phase, the mean fixation count was 19.4 (95% CI 18.6, 20.3) for agreements and 24.9 (95% CI 23.0–26.8) for disagreements ($P < 0.001$). In the XAI phase, these values were 22.6 (95% CI 21.8–23.4) for agreements and 28.7 (95% CI 26.4–31.4) for disagreements ($P < 0.001$, two-sided $t$-test, n_agreed = 355 cases, n_disagreed = 45 cases). Distributions of the fixation data can be found in Supplementary Fig. 2.

We also analyzed cases where the AI/XAI prediction was wrong but found no significant difference between the AI and XAI phases (Supplementary Fig. 3).

In summary, disagreements between dermatologists and classifier predictions significantly increased the number of ocular fixations in both AI and XAI phases across different study setups.

## Ocular fixations are correlated with dermatologists' experience levels

To assess the relationship between dermatologist experience levels and their ocular fixations in the AI and XAI phases, we performed a correlation analysis between the mean fixation count of each dermatologist and their experience in dermatology. Since we used mean fixation counts per dermatologist, outlier removal using the Interquartile Range (IQR) was conducted to ensure that the means accurately reflected their typical behavior. Dermatologists' experience levels were collected via the following experience brackets: less than 1 year, 1 to 3 years, 5 to 10 years, and over 10 years. Our analysis revealed a negative correlation of $-0.44$ (Spearman Correlation Coefficient; $P = 0.002$, $n = 46$ dermatologists) in the AI phase and $-0.31$ in the XAI phase (Spearman Correlation Coefficient; $P = 0.04$, $n = 46$ dermatologists) (Fig. 3b).

To explore these findings in more detail, we conducted separate analyses for the webcam-based study and the device-based study. In the webcam-based study, we found no significant correlations

($r = -0.40$, $P = 0.06$ with AI; $r = 0.37$, $P = 0.07$ with XAI, Spearman Correlation Coefficient; $n = 23$ dermatologists) between dermatologist experience and the number of fixations, while the device-based study showed a negative correlation of $-0.78$ (Spearman Correlation Coefficient; $P < 0.001$, $n = 23$ dermatologists) with AI and $-0.61$ (Spearman Correlation Coefficient; $P = 0.002$, $n = 23$ dermatologists) with XAI. For completeness, we have also provided the results obtained without the exclusion of outliers in Supplementary Table 2.

Thus, dermatologist experience was negatively correlated with ocular fixations during the AI phase and also during the XAI phase, with stronger correlations observed in the device-based study compared to the webcam-based study.

## Diagnostic disagreement between different dermatologists correlates with ocular fixations

To obtain insights into the relationship between diagnostic difficulty of the image and visual attention patterns, we assessed the change in the number of fixations as the difficulty of the lesion increased. To estimate diagnostic difficulty, we assigned a difficulty score to each image based on the amount of disagreement between the dermatologists. For this we calculated the entropy, which is a measure of impurity or randomness in a set of labels. Higher entropy indicated greater disagreement among dermatologists, and thus a higher difficulty score. Our findings revealed a correlation of 0.14 (Spearman Correlation Coefficient; $P < 0.001$, $n = 753$ images) between the number of fixations and diagnostic difficulty in the AI phase. However, no correlation was observed during the XAI phase ($r = 0.01$, $P = 0.76$, $n = 753$ images) (Fig. 3c).

To further understand these results, we analyzed the data from the webcam-based study and the device-based study separately. In the webcam-based study, we observed a correlation coefficient of 0.24 (Spearman Correlation Coefficient; $P < 0.001$, $n = 353$ images) between the number of fixations and diagnostic difficulty during the AI phase and a correlation of 0.13 (Spearman Correlation Coefficient; $P = 0.01$, $n = 353$ images) during the XAI phase. In the device-based study, we observed a correlation coefficient of 0.11 (Spearman Correlation Coefficient; $P = 0.02$, $n = 400$ images) between the number of fixations and diagnostic difficulty during the AI phase and a correlation of 0.13 (Spearman Correlation Coefficient; $P = 0.008$, $n = 400$ images) during the XAI phase. Distributions of diagnostic difficulty can be found in Supplementary Fig. 4.

**a**

**b**

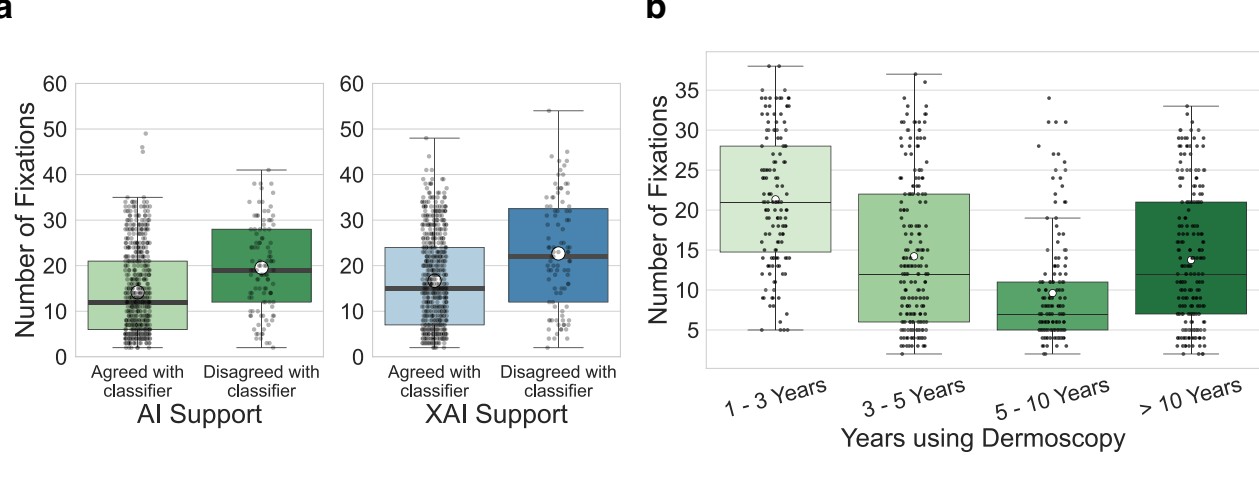

**c**

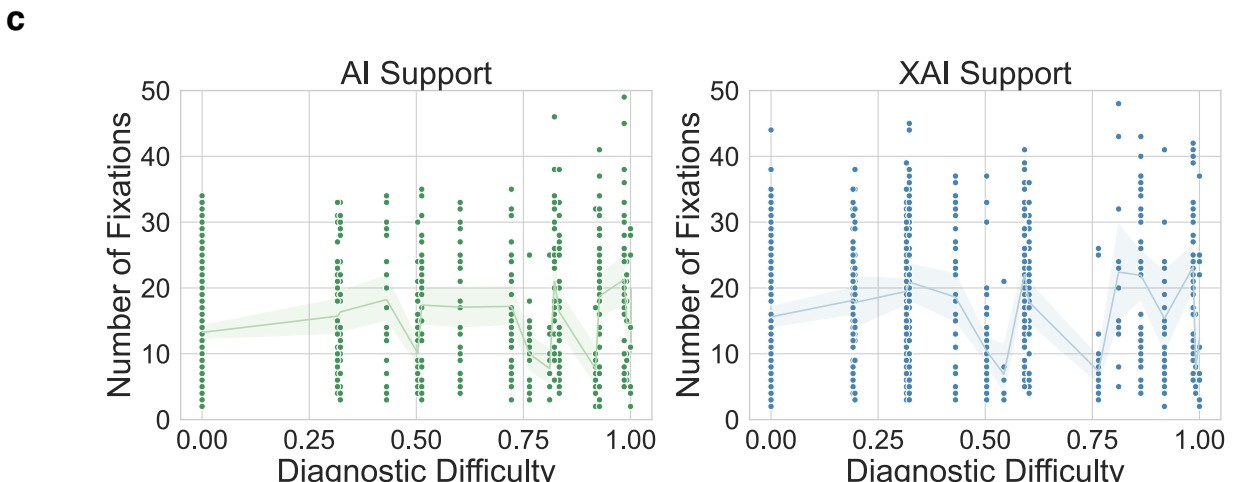

**Fig. 3 | Fixation patterns and cases of disagreement between dermatologist and classifier. a** Differences in fixation counts in cases where the dermatologist and classifier agreed (*P* < 0.001, two-sided *t*-test, n_agreed=316 cases, n_disagreed = 52 cases) and disagreed *(P* < 0.001, two-sided *t*-test, n_agreed = 317 cases, n_dis-agreed = 51 cases). The gray lines between the boxes connect the same dermatologist between the artificial intelligence (AI) and explainable artificial intelligence (XAI) phases, and the black lines connecting the boxes indicate the means across all dermatologists. The horizontal line on each box denotes the median value and the white dot denotes the mean. The upper and lower box limits denote the 1st and 3rd quartiles, respectively, and the whiskers extend from the box to 1.5 times the interquartile range. **b** Distributions of the number of fixations across different

experience levels. Fixations are negatively correlated with experience levels (two-sided Spearman Correlation Coefficient, *P* = 0.002, *n* = 61 dermatologists). The horizontal line on each box denotes the median value and the white dot denotes the mean. The upper and lower box limits denote the 1st and 3rd quartiles, respectively, and the whiskers extend from the box to 1.5 times the interquartile range. **c** Relationship between diagnostic difficulty and number of fixations. Difficult cases are associated with a higher number of fixations (two-sided Spearman Correlation Coefficient; *P* < 0.001, *n* = 753 images). Data are presented as mean values and bootstrapped confidence intervals derived from 1000 samples. Source data are provided as a Source Data file.

Thus, a slight correlation between the number of fixations and diagnostic difficulty was found in the AI phase, varying between webcam-based and device-based studies, but no consistent pattern was observed in the XAI phase.

## Discussion

Our work advances the understanding of the cognitive mechanisms in interaction of dermatologists with both AI and XAI in the context of melanoma diagnosis. We found that dermatologists are more accurate in their diagnoses when using XAI compared to plain AI. Additionally, we observed increased fixation counts while handling complex cases and when the predictions of the classifier diverge from the dermatologists' diagnoses.

We observed a statistically significant increase in dermatologists' diagnostic accuracy when supported by XAI compared to when being supported by plain AI. Our findings of a significant effect are partially in line with the trend observed in a study by Chanda et al.[2] which reported

a numerically higher but non-statistically significant increase with XAI over plain AI. A number of factors may account for this discrepancy. In their study, participants were required to complete a number of additional tasks, including the input of diagnostic confidence and trust in the classifier decision, in addition to lesion diagnosis. In comparison, our study consisted of a single task, namely lesion diagnosis. Our study also included a classifier that was slightly higher in terms of balanced accuracy (86.5% this study vs. 81% in Chanda et al.[2]). Additionally, our study included more precise information from the XAI, i.e., only the most confident explanation was presented. In contrast, all predicted explanations, including those with low classifier confidence, were presented in the study by *Chanda* et al.Such a large amount of information may have led to confusion and difficulty in interpretation. Furthermore, presenting the most confident explanation means that the explanation is more likely to be correct.

In the device-based study, where the diagnostic environment was more controlled, the balanced accuracy with XAI support was

numerically higher than with plain AI support. However, this improvement was not significant, which may be attributed to the smaller sample size ($n$ = 25). It is noteworthy that dermatologist accuracies were generally higher in the device-based study compared to the webcam-based study. This suggests that the controlled environment in which the device-based study was conducted may have provided a more conducive setting for accurate decision-making or a different selection of participants. The increase in diagnostic accuracy with XAI over plain AI could lead to enhanced patient outcomes, advancing dermatological care.

Eye-tracking analysis provided further insights into the cognitive processes underlying dermatologists' interactions with AI and XAI systems. Our results showed that the number of fixations was significantly higher when there was a disagreement between the dermatologist's diagnosis and the prediction of the classifier. This suggests that dermatologists spend more time and effort examining cases where there is a discrepancy, reflecting a deeper cognitive engagement with challenging cases. The higher fixation counts in these scenarios were observed in both the AI and XAI phases, indicating that the presence of explanations in XAI did not reduce the cognitive load but perhaps redirected it towards understanding the provided justifications. When encountering a classifier decision they disagree with, dermatologists might engage in a more in-depth analysis, revisiting specific details or searching for inconsistencies. As suggested by Kempt et al. dermatologists can leverage classifier predictions as second opinions to validate or reconsider their initial diagnoses, leading to more informed decision-making[31]. The relatively lower number of fixations when the dermatologist's diagnosis aligned with the prediction of the classifier suggests that the dermatologist might feel more confident and require less re-evaluation when their diagnosis is supported by the classifier. The agreement likely provides a form of validation that reduces the need for extensive additional examination. In practice, implementing AI interfaces that dynamically provide additional clarifications upon detecting user hesitation or prolonged fixations on specific lesion areas may be worth exploring.

Our findings also revealed discrepancies in absolute fixation numbers between the webcam-based and the device-based study. However, the relative trend remained consistent. For example, in both systems, fixation numbers were higher for disagreement compared to agreement (e.g., for AI: 8.2 vs. 19.2 in the webcam-based study and 13.4 vs. 24.9 in the device-based study; for XAI: 9.8 vs. 22.6 in the webcam-based study and 17.1 vs. 28.7 in the device-based study). This consistency in trends suggests that while absolute values differed due to variations in tracking accuracy, the comparative effects observed across conditions remained meaningful.

We found a negative correlation between fixation counts and dermatologist experience levels. This suggests that experienced dermatologists develop more efficient search patterns and require less time to visually inspect lesions compared to their less experienced colleagues. This efficiency is likely due to their familiarity and expertise in rapidly identifying key diagnostic features. This finding aligns with similar studies that show how experts develop efficient visual search strategies, leading to fewer fixations and shorter fixation duration[26,29]. However, different training backgrounds may also play a role in how dermatologists develop efficient fixation patterns, a factor not considered in this analysis. Training programs for dermatologists might benefit from incorporating visual fixation training, helping less experienced dermatologists develop more effective scanning techniques. Incorporating "eye movement modeling examples" where trainees view expert gaze patterns in real time or through replay can help novices internalize expert search strategies. We did not find any correlation between dermatologists' experience levels and their benefit with XAI over plain AI, in contrast to the positive correlation found in Chanda et al.[2]. In their study, experience was defined as their frequency of use and involvement in dermoscopy, i.e., rare use, occasional use,

regular use, regular use and involvement in science. In this study, we measured experience by their actual years of experience as dermatologists regardless of their frequency of use, which might explain the difference in findings between the two studies.

We found a positive correlation between the number of fixations and the diagnostic difficulty of the respective cases, suggesting that participants spent more time visually inspecting areas containing features that were challenging to diagnose. This aligns with the notion that increased cognitive load during visual tasks leads to more fixations and longer fixation durations[25].

One limitation of our study lies in the inherent drawbacks of webcam-based eye tracking systems, which often exhibit diminished reliability and accuracy compared to dedicated eye tracking devices, consequently generating data with reduced spatial precision. However, to mitigate this, we also used a dedicated eye tracking device to validate the results obtained from the webcam-based tracker. While this allowed us to cross-verify our findings, eye tracking technology cannot measure why a user looked at a certain element, as it provides objective, quantitative data but does not capture the subjective reasons behind visual attention[32]. Moreover, it is plausible that the interpretation of the images during the initial phase may have impacted the subsequent interpretation in the second phase. However, studies on visual recognition memory have shown that physicians' ability to recognize previously encountered medical images is limited[33–36]. Even though these studies are based on radiologic imaging rather than dermatology, it's likely that over time, dermatologists do not retain strong memory of medical images, and the interpretation of an image does not significantly influence their interpretation upon re-examination. Furthermore, our work does not resemble real-world clinical settings where the dermatologist has access to relevant patient metadata. Our findings may also not apply to real-world clinical settings where the majority of cases are clearly benign, as our study only included biopsy-verified lesions. Additionally, the potential influence of dermatologist diligence and attention levels during the phases on attained accuracy levels poses a limitation, where increased diligence may inflate or deflate accuracy on one or both of the phases independently of the AI system itself.

The findings of our study demonstrate the ability of XAI to enhance dermatologists' diagnostic accuracy and also enhance the understanding of the cognitive mechanisms involving dermatologists' interactions with AI when diagnosing melanoma. However, to conclusively establish the benefit of XAI over plain AI models in diagnosing melanoma, further prospective studies in real-world clinical environments should be conducted. We used fixation counts and fixation durations as objective measures to examine dermatologists' visual attention when interacting with AI-assisted diagnostic tools. However, fixation data alone cannot comprehensively capture the intricacies of clinical reasoning and diagnostic strategies. Therefore, future work may combine eye-tracking data with qualitative assessments, such as think-aloud protocols or questionnaires, to more fully understand the cognitive mechanisms of dermatologist-AI interactions. Future studies may also focus on the impact of XAI on cognitive load and trust in AI-assisted diagnosis.

## Methods
### Inclusion and ethics
Ethics approval was obtained from the ethics committee at the Technical University of Dresden (BO-EK-53012021), the Friedrich-Alexander University Erlangen-Nuremberg (69_21 Bc), the LMU Munich (21-0182), the University of Regensburg (20-2190-103), the Julius-Maximilians University Wuerzburg (293/20_z) and from the University Hospitals Mannheim (2020-656N) and Essen (20-9784-BO). Informed consent was collected from all participants. We did not collect any data on sex and gender of the clinicians participating in our reader study. As compensation, we offered them the opportunity to be credited as a collaborator of our work. The dermoscopic skin lesion image used in

Fig. 1 is part of the open-source HAM10k dataset, which is already published (reference cited).

## Datasets

In our work, we utilized dermoscopic skin lesion images of melanomas and nevi from the HAM10000[37] and a multi-clinic prospectively-collected dataset. To minimize label noise, we selected only the biopsy-verified lesions from HAM10000 ($n = 1981$ unique lesions). The prospectively-collected dataset ($n = 1654$ unique lesions) consisted entirely of biopsy-verified lesions. Since both datasets contained multiple images of the same lesion, we randomly selected only one image per lesion and excluded the rest. Approximately 22% of the lesions in the entire dataset were melanomas and 78% were nevi.

The entire dataset was randomly divided into training (80%), validation (10%), and test sets (10%). To further assess generalizability, we incorporated an external test set comprising images from a single clinic within the prospectively-collected dataset, ensuring these images were excluded from the other sets. We randomly selected 48 images (3 groups with 16 images each) from the test set for the reader study. Each group consisted of 8 melanomas and 8 nevi.

## Participants

We recruited dermatologists with varying levels of experience ranging from assistant dermatologists to clinic directors. Invitations were sent via email through our collaboration network, utilizing public contact information from the International Society for Dermoscopy website and university clinic webpages. Additionally, we included dermatologists from private clinics. Participant numbers and flow is illustrated in Supplementary Fig. 5.

## Classifier

We adapted the explainable classifier introduced by Chanda et al.[2], which explains its decisions using established visual characteristics. However, its generalization performance was insufficient due to reliance on annotated training data. To address this, we modified the classifier to learn from both annotated and unannotated images.

We added two prediction heads to the output layer: a diagnosis prediction head and a characteristics prediction head. For images without annotated characteristics, only the diagnosis loss is optimized. For images with annotated characteristics, we optimized the diagnosis loss, the characteristics loss, and the attention loss defined in Chanda et al.[2]. This approach increased the amount of training data and improved generalization performance.

## Study design and eye tracking

Our study, conducted in two phases between December 2023 and June 2024, involved monitoring the eye movements of participants using a web-based eye-tracking software. Additionally, a separate set of participants had their eye movements tracked using a dedicated eye-tracking device. The study included three groups of participants, with each group reviewing 16 mutually exclusive dermoscopic images (8 melanomas and 8 nevi). This image limit was based on feedback from a pilot study, which indicated that more than 20 images led to increased participant fatigue, so 16 images per group were selected to maintain optimal engagement and accuracy.

## Web-based study

Initially, we used the web-based software realeye.io version 9.0, which tracks eye movements using a webcam video feed. Participants were first required to complete a calibration step to ensure accurate tracking. Since webcam-based eye tracking is sensitive to lighting conditions, they were instructed to keep a light source in front of them and remove any light sources behind them. They were also asked to keep their head position fixed during the session. The software automatically pauses the session and alerts the user if their head position deviates. A fixation was recorded whenever a participant's eyes focused on a specific point for 100–300 milliseconds, representing a clear point of attention. The collected data included time-stamped coordinates indicating where on the screen the participant's attention was directed.

**AI Phase.** In this phase, participants were asked to diagnose 16 dermoscopic images of melanomas and nevi, supported by an AI system that provided predictions for each image (nevus or melanoma) (Fig. 1b). They received both the images and the AI diagnoses simultaneously. They received instructions on setting up the study, including the required calibration steps. The distribution of melanomas and nevi was not disclosed to them. Participants were asked to complete the task within two weeks. We randomly divided the participants into three groups, with each group receiving 16 random images (8 melanomas and 8 nevi) from the test set. The image sets for each group were mutually exclusive to ensure a broad coverage of images. Participants were informed that the task would consume approximately 10 to 12 min to complete. We did not set an upper limit on completion time for exclusion, as certain complex cases might require more time to diagnose. Participants were allowed to pause and resume their work, so a longer completion time did not necessarily indicate insincere efforts, although this did not happen. Individuals who withdrew in the middle of the study were excluded from the analyses. 53 dermatologists participated in this phase and completed the task.

**XAI Phase.** In the XAI phase, we incorporated the 53 participants who successfully concluded the AI phase. In this phase, the participants were asked to diagnose the same 16 dermoscopic images of melanomas and nevi. They were supported by an XAI system that provided predictions for each image, as well as explanations for the predictions (Fig. 1c). They received both the images and the XAI diagnoses and explanations simultaneously. The explanations consisted of the characteristics that are relevant in diagnosing melanoma/nevus including polygon-based region indications of the detected characteristics. We ensured a minimum two-week interval between completing the AI phase and initiating the XAI phase, and we did not disclose that these were the same lesions from the preceding phase. The average interval period was 3 weeks. Similar to the AI phase, participants were instructed to complete the task within a two-week timeframe. Participants were presented with images from the AI phase in the same sequence, along with the diagnosis of the AI for the respective lesion (nevus or melanoma) and its explanation for the prediction. Participants were informed that the task would consume approximately 10 to 12 min to complete. Three participants failed to complete this phase within the stipulated deadline, resulting in a total of 50 participants.

**Onsite validation.** To validate the results of the web-based study, we conducted an additional onsite validation study using a dedicated eye-tracking device (Pupil Labs Core). This setting was identical to the previous AI and XAI phases but incorporated the use of a dedicated eye-tracking device instead of webcam-based eye tracking. A total of 25 dermatologists participated in this onsite validation study. They were asked to complete both the AI and XAI tasks while wearing the dedicated device.

The protocol for this phase mirrored the web-based study, including the 16 dermoscopic images (8 melanomas and 8 nevi), and the presentation of the diagnosis of the AI in the AI phase and the diagnosis and explanations of the AI in the XAI phase. Similar to the web-based study, we ensured a minimum of two weeks between the AI and XAI phases.

The onsite validation phase was necessary to address concerns and potential limitations associated with the web-based eye-tracking study. First, the precision and accuracy of webcam-based eye tracking can be significantly lower than that of dedicated eye tracking devices. Webcams are more susceptible to variations in lighting conditions,

user positioning, and other environmental factors, which can introduce noise and reduce the quality of the collected data. By using a dedicated device in a controlled onsite setting, we aimed to rule out these potential sources of error and ensure the robustness of our findings. Another concern was the potential for webcam-related inconsistencies. Different webcam models used by the participants might have influenced the performance of the web-based eye-tracking software, leading to variability in data quality. The onsite validation using a standardized device provided a consistent and controlled environment, allowing us to verify that the eye-tracking data was reliable and not confounded by these variables.

## Software
All code was written in Python (3.9.9). PyTorch (1.10.0), PyTorch Lightning (1.5.10), Albumentations (1.0.3), NumPy (1.22.2), Pandas (1.4.0), SciPy (1.8.0), OpenCV (4.5.5), Scikit-learn (1.1.0), Matplotlib (3.1.1), and Seaborn (0.11.2) were used for image processing, model development and training, data analysis, and visualization.

## Statistics and reproducibility
The primary endpoint was to compare the dermatologists employing AI and XAI with respect to their balanced accuracy scores. All pairwise significance testing was performed using the two-sided paired $t$-test. To calculate confidence intervals, we utilized the bootstrapping method with 10,000 samples and a random seed of 42 each time the confidence interval was calculated. No formal statistical methods were used to estimate the minimum sample size for the study. Instead, the approach was to collect as many samples as possible to maximize the robustness of the analysis. A sample size greater than 25 was considered sufficient based on general statistical guidelines for achieving approximate normality in many parametric tests. The number of dermatologists (53 in the web-based study, 25 in onsite validation) was driven by recruitment feasibility and alignment with similar reader studies in medical AI evaluation. Blinding was not relevant to our study because the investigators neither provided diagnoses nor influenced them. The participants were not aware of the existence of grouping. The diagnoses were made entirely by participants, and all eye-tracking data were recorded automatically, minimizing any potential bias from investigator knowledge of group allocation.

## Reporting summary
Further information on research design is available in the Nature Portfolio Reporting Summary linked to this article.

## Data availability
The data generated in our study, which includes the pseudonymized reader study data and the fixations data are accessible on Figshare: https://figshare.com/s/5f0b0f18c20f0a850dc7. The HAM10000 dataset[37] is publicly available and can be accessed here: https://dataverse.harvard.edu/dataset.xhtml?persistentId=doi:10.7910/DVN/DBW86T. External research projects may request access to the prospectively-collected additional dataset used in our study, specifically for the purpose of advancing skin (cancer) research. Access is granted following an application and approval process managed by the SCP Data Protection Committee, which evaluates requests based on criteria such as alignment with patient consent, a valid ethics vote, and other relevant requirements (i.e., non-commercial (skin) cancer research). All remaining data is publicly available. Commercial use of the data is prohibited. All remaining data supporting this work are available in the main article, supplementary information, or source data file. Source data are provided with this paper.

## Code availability
The custom code developed in this work is accessible at https://github.com/DBO-DKFZ/EyeTracking[38].

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

## Acknowledgements

We sincerely thank the dermatologists who participated in the reader study for their time and expertise. Their valuable contributions were essential to this work. Federal Ministry of Health, Berlin, Germany, grant number 2520DAT801, T.J.B. Ministry of Social Affairs, Health and Integration of the Federal State Baden-Württemberg, Germany, grant number 53–5400.1-007/5, T.J.B.

## Author contributions

T.J.B., T.C., S.H., and T.C.B. conceived of and designed the overall study. T.J.B. and T.C. were responsible for reader study participant recruitment. T.J.B. acquired funding and resources. T.C. conducted the reader study and data collection supported by T.J.B. T.C. performed the analyses with support from T.H.L. T.J.B. supervised the whole project and takes responsibility for the published data as well as all analyses conducted. T.C. drafted the initial version of the manuscript. H.K., P.T., M.V.H., C.B., J.S.U., B.S., C.B., C.N.D., M.G., J.N.K., C.V.S., B.D., H.D., M.J., Ju.W., Jo.W. provided clinical and/or machine learning expertise and contributed to the interpretation of the results and critically revised the manuscript.

## Funding

## Competing interests

J.N.K. declares consulting services for Bioptimus, France; Owkin, France; DoMore Diagnostics, Norway; Panakeia, UK; AstraZeneca, UK; Scailyte, Switzerland; Mindpeak, Germany; and MultiplexDx, Slovakia. Furthermore he holds shares in StratifAI GmbH, Germany, has received a research grant by GSK, and has received honoraria by AstraZeneca, Bayer, Daiichi Sankyo, Eisai, Janssen, MSD, BMS, Roche, Pfizer and Fresenius. T.J.B. would like to disclose that he owns a software company (Smart Health Heidelberg GmbH, Handschuhsheimer Landstr. 9/1, 69120 Heidelberg), outside of the scope of the submitted work. No other competing interests are declared by any of the authors.

## Additional information

[1]Digital Biomarkers for Oncology Group, German Cancer Research Center (DKFZ), Heidelberg, Germany. [2]Department of Biostatistics, German Cancer Research Center (DKFZ), Heidelberg, Germany. [3]Department of Dermatology, Medical University of Vienna, Vienna, Austria. [4]Department of Dermatology, Uniklinikum Erlangen, Friedrich-Alexander-Universität Erlangen-Nürnberg, Erlangen, Germany. [5]Skin Cancer Unit, German Cancer Research Center (DKFZ), Heidelberg, Germany. [6]Department of Dermatology, Venereology and Allergology, University Medical Center Mannheim, Mannheim, Germany. [7]DKFZ Hector Cancer Institute at the University Medical Center Mannheim, Mannheim, Germany. [8]Department of Dermatology, Venereology and Allergology, University Hospital Frankfurt, Goethe-University Frankfurt, Frankfurt, Germany. [9]Department of Dermatology, Escuela de Medicina, Pontificia Universidad Católica de Chile, Santiago, Chile. [10]Department of Dermatology, Venereology and Allergology, University Hospital Würzburg, Würzburg, Germany. [11]Else Kroener Fresenius Center for Digital Health, Faculty of Medicine, Dresden, Germany. [12]University Hospital Carl Gustav Carus, TUD Dresden University of Technology, Dresden, Germany. [13]Department of Internal Medicine, University Hospital Aachen, RWTH University of Aachen, Aachen, Germany. [14]Dres. Durani, Outpatient Clinic for Dermatology, Heidelberg, Germany. [15]Dr. Martin Jansen, Outpatient Clinic for Dermatology, Heidelberg, Germany. [16]Dres. Wacker, Outpatient Clinic for Dermatology, Heidelberg, Germany. ✉e-mail: titus.brinker@dkfz.de

## Reader Study Consortium

Nina Booken[17], Verena Ahlgrimm-Siess[18], Julia Welzel[19], Oana-Diana Persa[20], Florentia Dimitriou[21], Stephan Alexander Braun[22], Lara Valeska Maul[23], Antonia Reimer-Taschenbrecker[24,25], Sandra Schuh[19], Falk G. Bechara[26], Laurence Feldmeyer[27], Beda Mühleisen[28], Elisabeth Gössinger[29], Stephan Alexander Braun[30,31], Van Anh Nguyen[32], Julia-Tatjana Maul[33], Friederike Hoffmann[34], Claudia Pföhler[35], Janis Thamm[19], Wiebke Ludwig-Peitsch[36], Daniela Hartmann[37], Laura Garzona-Navas[38], Martyna Sławińska[39], Panagiota Theofilogiannakou[40], Ana Sanader Vucemilovic[41], Juan José Lluch-Galcerá[42], Aude Beyens[43,44], Dilara Ilhan Erdil[45], Rym Afiouni[46], Vanda Bondare-Ansberga[47], Martha Alejandra Morales-Sánchez[48], Arzu Ferhatosmanoğlu[49], Roque Rafael Oliveira Neto[50], Lidija Petrovska[51], Amalia Tsakiri[52], Hülya Cenk[53], Sharon Hudson[54], Miroslav Dragolov[55], Zorica Zafirovik[56], Ivana Jocic[57], Alise Balcere[58], Zsuzsanna Lengyel[59], Alexander Salava[60], Isabelle Hoorens[43], Sonia Rodriguez Saa[61], Emőke Rácz[62], Gabriel Salerni[63], Karen Manuelyan[64], Amr Mohammad Ammar[65], Michael Erdmann[66], Nicola Wagner[66], Jannik Sambale[66], Stephan Kemenes[66], Moritz Ronicke[66], Lukas Sollfrank[66], Caroline Bosch-Voskens[66], Ioannis Sagonas[66], Thomas Breakell[66], Christopher Uebel[66], Lisa Zieringer[66], Michael Hoener[66], Leonie Rabe[67], Tim Sackmann[67], Julia Baumert[67], Marthe Lisa Schaarschmidt[67], Nadia Ninosu[67], Kaan Yilmaz[67], Danai Dionysia[67], Franca Christ[67], Sarah Fahimi[67], Sabina Loos[16], Ani Sachweizer[16], Janika Gosmann[8], Tobias Weberschock[8], Ufuk Erdogdu[8], Amelie Buchinger[8], Jasmin Lunderstedt[8], Timo Funk[8], Hess Klifo[8], Sebastian Kiefer[8], Dietlein Klifo[8] & Malin Kalski[8]

[17]University Medical Center Hamburg-Eppendorf, UKE Department of Dermatology and Venereology, Hamburg, Germany. [18]Universitätsklinik für Dermatologie und Allergologie, Universitätsklinikum Salzburg, Salzburg, Austria. [19]Klinik für Dermatologie und Allergologie, Universitätsklinikum Augsburg, Augsburg, Germany. [20]Department of Dermatology and Allergology, Technical University of Munich, Munich, Germany. [21]Department of Dermatology, University Hospital of Zurich, Zurich, Switzerland. [22]Department of Dermatology, University Hospital Münster, Münster, Germany. [23]Klinik für Dermatologie, Universitätsspital Basel, Basel, Switzerland. [24]Northwestern University, Feinberg School of Medicine, Department of Dermatology, Chicago, IL, USA. [25]Medical Center – University of Freiburg, Department of Dermatology, Freiburg, Germany. [26]Klinik für Dermatologie, Venerologie und Allergologie, Ruhr-Universität Bochum, Bochum, Germany. [27]Department of Dermatology and Allergology, Cantonal Hospital Lucerne, Lucerne, Switzerland. [28]Hautartz Zentrum, Liestal, Switzerland. [29]Department of Dermatology, University Hospital of Basel, Basel, Switzerland. [30]Department of Dermatology, University Hospital Muenster, Muenster, Germany. [31]Department of Dermatology, Medical Faculty, Heinrich-Heine University, Duesseldorf, Germany. [32]Universitätsklinik für Dermatologie, Venerologie und Allergologie, Innsbruck, Austria. [33]Department of Dermatology, University Hospital Zurich, Zurich, Switzerland. [34]Department of Dermatology and Allergy, University Medical Center Bonn, Bonn, Germany. [35]Universitätsklinikum des Saarlandes, Klinik für Dermatologie, Homburg/Saar, Germany. [36]Klinik für Dermatologie und Phlebologie, Vivantes Klinikum im Friedrichshain, Berlin, Germany. [37]Klinik für Dermatologie und Allergologie, München Klinik Thalkirchner Straße, München, Germany. [38]Hospital Clinica Biblica, San José, Costa Rica. [39]Department of Dermatology, Venereology and Allergology, Faculty of Medicine, Medical University of Gdańsk, Gdańsk, Poland. [40]Dermatology Department, Evaggelismos General Hospital of Athens, Athens, Greece. [41]Clinical Hospital Centre Split, Split, Croatia. [42]Dermatology Department, Hospital Universitari Germans Trias i Pujol, Barcelona, Spain. [43]Department of Dermatology, Ghent University Hospital, Ghent, Belgium. [44]Center for Medical Genetics Ghent, Ghent University Hospital, Ghent, Belgium. [45]Istanbul Teaching and Research Hospital, Istanbul, Turkey. [46]Faculty of Medicine, Saint-Joseph University, Beirut, Lebanon. [47]Rîga 1st Hospital, Riga, Latvia. [48]Centro Dermatológico Dr. Ladislao de la Pascua, Mexico City, Mexico. [49]Department of Dermatology and Venerology, Karadeniz Technical University Faculty of Medicine, Istanbul, Turkey. [50]Federal University of Mato Grosso UFMT, Cuiabá, Brazil. [51]Dermatology Department, PHI Clinical Hospital Shtip, Shtip, North Macedonia. [52]Private Clinic, Thessaloniki, Greece. [53]Department of Dermatology, Pamukkale University, Denizli, Turkey. [54]Clinical Nurse Consultant, Melbourne, Australia. [55]Mirabel Clinic, Burgas, Bulgaria. [56]University Clinic of Dermatology, Faculty of Medicine, University "Ss. Cyril and Methodius", Skopje, North Macedonia. [57]Department of Dermatology and Venereology, School of Medicine, Military Medical Academy, Belgrade, Serbia. [58]Riga Stradins University Department of Dermatology and Venereology, Riga, Latvia. [59]Department of Dermatology, Venerology and Oncodermatology, University of Pécs, Pécs, Hungary. [60]Department of Dermatology and Allergology, Helsinki University Hospital, Helsinki, Finland. [61]Dermatology Department, Hospital El Carmen, Mendoza, Argentina. [62]Department of Dermatology, University Medical Center Groningen, University of Groningen, Groningen, the Netherlands. [63]Dermatology Department, Hospital Provincial del Centenario de Rosario, Universidad Nacional de Rosario, Rosario, Argentina. [64]Department of Dermatology and Venereology, Clinic of Dermatology and Venereology, University Hospital, Trakia University, Stara Zagora, Bulgaria. [65]Al-Azhar University, Cairo, Egypt. [66]Department of Dermatology, Uniklinikum Erlangen, Friedrich-Alexander-University Erlangen-Nürnberg, Erlangen, Germany. [67]Department of Dermatology, Venereology and Allergology, University Medical Center Mannheim, Ruprecht-Karl University of Heidelberg, Mannheim, Germany.

