## [Transparent Peer Review file · Nature Communications]

Dermatologist-like explainable AI enhances melanoma diagnosis accuracy: eye-tracking study

Corresponding Author: Dr Titus Brinker

Version 0:

Reviewer comments:

Reviewer #1

(Remarks to the Author)

Thank you for the opportunity to review this manuscript, entitled "Dermatologist-like Explainable AI Enhances Accuracy in Diagnosing Melanoma: An Eye-Tracking Study." Overall, the authors have chosen a novel path for assessing human-AI interaction, which adds significant value to the field of dermatology.

After reviewing this manuscript, I wish to provide a few thoughts to consider as this manuscript moves forward here or elsewhere.

- The study benefits from a well-structured, two-phase design comparing AI and XAI systems. The validation study using a dedicated eye-tracker enhances the reliability of the findings, particularly addressing concerns about potential biases in web-based data collection.
 - The statistically significant improvement in diagnostic accuracy when using XAI over AI alone is a compelling outcome. The study's contribution to enhancing trust and decision-making in AI-driven tools for melanoma detection is clear and addresses an important clinical challenge.
 - While the study discusses increased fixation counts in cases of disagreement between dermatologists and AI, further elaboration on how cognitive load specifically interacts with trust in XAI systems would strengthen the interpretation. More detail on how the XAI explanations alleviate or redirect cognitive load would be valuable.
 - The dataset primarily consists of biopsy-verified images, which ensures high-quality input. However, the study could benefit from discussing how these results generalize to broader clinical settings where input data might not always be as clean. Consider including limitations regarding this in the discussion.
 - While there is a brief mention of experience levels, a more detailed exploration of how dermatologist experience influenced their interaction with XAI (beyond fixations) would enrich the findings. Breaking down diagnostic accuracy improvements across experience brackets could provide more actionable insights for clinical implementation.
 - The study employs both webcam-based and dedicated eye-tracking methods, which is commendable for cross-validation. However, webcam-based eye tracking is inherently less precise due to variations in participants' environments (e.g., lighting, seating position) and the lower resolution of consumer webcams. These factors can lead to less accurate fixation data, which could introduce noise or error in the interpretation of results. It may bear discussing this as a limitation. Additionally, how much discrepancy exists between the webcam and dedicated systems? Reporting metrics such as fixation accuracy, calibration reliability, or the number of discarded data points from the webcam-based system would enhance transparency.
 - Understandably, the authors interpret longer fixation durations as indicative of cognitive load and diagnostic difficulty. However, fixation duration can also be influenced by other contextual factors, such as distraction, unfamiliarity with the interface, confusion, or interest in specific visual features. The authors do not account for these alternative explanations, but it may bear mentioning the contextual nature of fixations in the limitations.
 - The manuscript mentions that expert dermatologists exhibit fewer fixations, which is consistent with the literature on visual search patterns among experts. However, further comparison between novice and expert behavior could be useful pertaining to their interactions with the XAI explanations? Are experts more likely to ignore AI predictions, or do they verify them in a different manner than novices? Perhaps these are best answered in a future study, but they are compelling questions that are not answered in the current manuscript.
- These are the main thoughts that I had as I reviewed this manuscript. I applaud the authors for taking an innovative approach to examining human-AI interactions in dermatological studies, and I wish them the best of luck moving forward.

(Remarks on code availability)

Reviewer #2

(Remarks to the Author)

The authors report on an eye-tracking study that compares how using AI and XAI systems, resp., influences the diagnostic accuracy of dermatologists when diagnosing pigmented skin lesions. The study is well thought out, well executed, and also reported well. Nevertheless, there are a number of possible improvements, both big and small, that could be made to the manuscript. I will go through these possible improvements in the order in which they appear in the manuscript, at times indicating whether I consider these suggestions to be of major or minor importance. I am thus sorry for not providing a better, more consistent narrative in this review.

First, in the abstract, the authors claim that "[t]hese insights have significant implications for the design of AI tools for visual tasks in dermatology", but do not follow up in any significant manner (pun intended) by pointing out what these implications are (there is only a vague mention of "[t]raining programs for dermatologists might benefit from incorporating visual fixation training"---unclear what this may be---in the discussion). Please improve.

Mainly in the introduction, and less so throughout the manuscript, there are sentences that seem overly constructed to the point of either being obvious or meaningless. One such example is "A diagnosis assistance system requires local, strongly end-user-focussed explanations as dermatologists need to assess the quality of the machine suggestions on a case-by-case level". It may be nit-picking, but I encourage the authors to put themselves in the readers' shoes: What is a "local explanation"? It is a given that an explanation in a visual discipline such as dermatology will have to point out characteristic features of a lesion, most of which are local. So I assume this means pointing out the presence of local features. What is "end-user-focussed [sic]"? Directed at the user? Who else would it be directed at? "assess the quality [...] on a case-by-case level": How else to assess than at the level of the individual lesions?

Another example, which is very minor, but should point out the need to go over the manuscript with a fine-toothed comb: "Their work, however, analyzed pathologists and AI independently, rather than in tandem. Consequently, eye tracking technology can precisely capture where and for how long dermatologists focus their visual attention while using XAI systems." How is the second sentence related to the first via "consequently"?

The authors set out to "[gain] insights into the visual patterns and diagnostic strategies employed by dermatologists when utilizing AI", which is a very interesting goal. However, they "only" (not to diminish their contribution) report on number and duration of eye tracking fixations, which is (arguably) an entirely different thing.

The authors report that "[t]he explanations consisted of the characteristics that are relevant in diagnosing melanoma/nevus including polygon-based region indications of the detected characteristics", but the caption of Fig 1c gives an explanation of pseudopods that is inconsistent with the polygon that marks the central region of the lesion. Please clarify.

The authors refer to "two devices: one with a webcam-based eye tracker and another with a dedicated eye tracker". Later on, the term "device" as in "device-based" is reserved for the dedicated eye tracker. This is confusing at first, so please aim for more consistent terminology.

This is major: I am not sure whether the statistical analysis by using a t-test is correct, when in essence the comparison is between two proportions. Please refer to the paper "L.D. Nelson and J.P. Simmons, Moniker maladies: When names sabotage success, *Psychol. Sci.* 18(12) (2007), pp. 1106–1112." and its subsequent debunking in "B.D. McCullough and T.P. McWilliams, Baseball players with the initial "K" do not strike out more often, *J. Applied Statistics* 37(6) (2010), pp. 881–891" for a thorough analysis of an instance in which the wrong test was used.

Please explain why there are only discrete balanced accuracy levels in Fig. 2a. I assume it is for clarity of presentation, but it should be mentioned nonetheless.

There is one detail of the diagnostic process missing, or I might have overlooked it: Do the physicians first and independently diagnose the lesion, and later get the (X)AI diagnosis, or do they see the (X)AI diagnosis before making their own diagnosis?

This is major: The authors want to validate webcam-based fixation numbers by later also using a dedicated eye-tracking device. This seems like a good idea to me. When the numbers indicate that outputs of the two systems do not align at all (8.2 vs. 19.2 for AI/agreement, 13.4 vs. 24.9 for AI/disagreement, 9.8 vs. 22.6 for XAI/agreement, 17.1 vs. 28.7 for XAI/disagreement), there is little to no mention of this in the discussion, when it seems that this observation alone may invalidate most of the findings in the webcam-based part of the study.

"Diagnostic difficulty" and "label difficulty" are used interchangeably. Please use the same term of the same concept.

In the methods: "Using biopsy-verified images ensured that the images in the work were sufficiently challenging for diagnostic purposes"---what does having a biopsy-verified diagnosis available or not have to do with the difficulty of diagnosing a lesion"? Please clarify or rephrase.

There seem to be conflicting numbers as to the distribution of the lesion classes (benign/malignant) in the study: 22% vs. 78% (p. 14), 8 melanomas and 8 nevi (p. 15). Please clarify.

"require more annotation time" (p. 15): The rest of the manuscript refers to diagnosis, it is unclear at which stage annotation of lesions take place. Furthermore, I do not see results of this annotation in the manuscript. Please clarify.

The last two paragraphs on p. 16 contain redundant explanations, please tighten the exposition.

Again, this is major (but same point as above): The authors state that "[t]herefore, the onsite validation was critical to confirm that the observed eye-tracking patterns were genuinely reflective of the dermatologists' visual attention and decision-making processes, rather than artifacts of the web-based methodology." The numbers seem to indicate that some results really are "artifacts of the web-based methodology". A possible revision of the manuscript needs to focus on getting this point right.

(Remarks on code availability)

Reviewer #3

(Remarks to the Author)

1. Key Results

This paper presents a two-phase sequential reader study involving 76 dermatologists evaluating 16 pigmented lesions each to elucidate how dermatologists engage with both AI alone (phase 1) and XAI (phase 2) diagnostic support. Eye-tracking technologies using both webcams (n=51) and device-based (n=25) were utilized to objectively quantify engagement with both AI systems. The key results were that balanced accuracy improved slightly with XAI over AI alone by 2.8% and was not correlated with clinician experience, ocular fixations increased with clinician disagreement with AI/XAI and for more diagnostically difficult lesions, and ocular fixations were negatively correlated with dermatologist experience.

2. Significance

The most notable contribution of this paper relates to the evaluation of the AI support systems' impact on clinician cognitive load as inferred through ocular fixations. A strength is the robust in person validation with a device-based technology, the results of which were largely directional with the results of the larger webcam-based cohort. The increased ocular fixations that occur with clinician disagreement with AI is interesting, including the related finding that XAI does not reduce cognitive load and instead seems to facilitate increased clinician cognitive engagement during disagreements relative to cases of agreement.

The finding of statistically significant improved diagnostic accuracy with XAI over AI is more challenging to interpret in light of the prior well done study by Chanda et al. (Ref#2 Nat Commun. 2024 Jan 15;15(1):524) that found in a larger study with >100 dermatologists that XAI did not increase diagnostic accuracy over AI alone. The authors provide very reasonable considerations for why this present result may have differed with an improved classifier and different experimental design. However, given the relatively small magnitude improvement in accuracy observed, it does seem like the larger question of whether XAI will be advantageous in a clinically impactful manner for diagnostic accuracy over identically well performing AI alone is not entirely resolved, at least for this specific task of classifying melanomas from nevi.

3. Data and Methodology

- Figure 3a. The figure legend states: "The gray lines between the boxes connect the same dermatologist between the AI and XAI phases, and the black lines connecting the boxes indicate the means across all dermatologists." I do not see any of these lines in this version of the figure in the manuscript

- I appreciate that the authors acknowledge as a potential limitation in the Discussion the plausibility that 'interpretation of the images during the initial phase may have impacted the subsequent interpretation in the second phase.' This was one of my concerns with the study design, as beyond the addition of XAI, there is a theoretical confounding factor of the clinicians essentially getting a second pass at evaluating the same case within a relatively short timeframe and could apply additive scrutiny to identify some key diagnostic feature initially missed, regardless of the influence of the XAI. I agree that the references 34-37 on the limitations of expert visual memory help to increase confidence in the conclusions and would suggest moving these from Methods to the relevant portion in discussion. At the same time, these references seem to be based on radiology and cytology expert memory studies rather than anything related to dermatologic lesion assessment, and should be more accurately qualified as such as they do not technically provide direct evidence regarding dermatologists' ability to recall skin lesions as is suggested in the manuscript

- Related to this, for the on-site device-based validation, was there a similar >2 week interval between the AI and XAI phases as the webcam based groups? I could not tell from the description in the Methods section

4. Analytical Approach

The statistical testing appeared appropriate to me

5. Suggested Improvements

- An interesting consideration of explainable AI is whether it engenders trust that could then lead clinicians to accept faulty recommendations that they may have otherwise rejected with an AI without explainability. If possible, would be interested in the subset of cases where the classifier provided a classification discordant with the ground truth ('faulty' recommendation), and whether clinicians did better with AI or XAI support in this subset of cases. Further, would be interesting to see what the effect was on number of fixations for both AI alone and XAI for this subset.

6. Clarity and Context

- The relevant prior work in XAI and eye tracking studies is well written and concisely summarized
- The lack of correlation between dermatologist experience and increase in diagnostic accuracy with XAI over AI was notable as related prior work had previously identified a correlation between dermoscopy experience and increase in accuracy with XAI over AI (Ref#2 Chanda et al.). The present result suggests that there may be some other clinician factor(s) beyond dermoscopic expertise that might predict which clinicians may derive more benefit in diagnostic accuracy from XAI over AI alone. While it is beyond the scope of this present study design, it would be informative if the authors might be willing to speculate on what other clinician-specific factors beyond dermoscopic experience might be influencing who benefits "more" from XAI in terms of improved diagnostic accuracy (? Training background, baseline attitudes towards AI, etc), and how this might be explored in future work
- minor suggestion for clarity: The last paragraph of Results that begins "In summary, our analysis reveals that disagreements between dermatologists and classifier predictions correlate with increased ocular fixations..." is confusing in that it is presented within the "Diagnostic disagreement..." subsection but actually summarizes the entire Results section. This could easily be addressed in copy editing with appropriate spacing, or if not possible consider removal as all of the key results are addressed in the Discussion anyway

(Remarks on code availability)

Version 1:

Reviewer comments:

Reviewer #1

(Remarks to the Author)

The authors have thoroughly and effectively addressed all concerns raised in my initial review. They provided clear explanations and revisions that strengthen the manuscript's clarity, methodology, and presentation of findings.

With the interaction between cognitive load and trust in explainable AI (XAI), the authors acknowledged the relevance of this relationship but clarified that trust was not measured in their study. They appropriately framed this as an area for future research and emphasized their primary focus on fixation counts as indicators of cognitive effort. Similarly, they addressed concerns about generalizability by acknowledging that their dataset of biopsy-verified images is particularly relevant to diagnostically challenging cases but may not fully represent routine clinical settings. A corresponding statement has been added to the limitations section.

The influence of dermatologist experience on XAI use was also investigated, and while no significant differences were found, the authors explained this in light of differing definitions of experience across studies. They incorporated this discussion into the revised manuscript, strengthening their justification. Additionally, the concerns regarding webcam-based eye-tracking precision were addressed. The authors expanded the Methods section to describe how variability was controlled and provided supplementary data on exclusion criteria and discrepancies between webcam and dedicated device tracking. They clarified that while absolute fixation numbers differed between tracking methods, the relative trends remained consistent, reinforcing the validity of their findings.

Several clarifications were made regarding the interpretation of fixation data. The authors acknowledged that fixation duration may be influenced by factors beyond cognitive load and diagnostic difficulty, such as interface familiarity. They also explained how distractions were accounted for in the study. In response to my comments on expert versus novice interaction with XAI predictions, they recognized this as an important question but explained that their dataset was not designed to answer it. They appropriately positioned this as a future research avenue.

Concerns about the use of t-tests in the statistical analysis were addressed with a well-reasoned defense. The authors clarified that their comparison involved continuous fixation count data rather than proportions, making the t-test a valid choice. This explanation was incorporated into the Methods section. Additionally, several key revisions were made to clarify the AI/XAI decision process, improve figure descriptions, and ensure consistency in terminology. Discrepancies in fixation counts between tracking methods were explicitly discussed, reinforcing the reliability of their conclusions.

Finally, the authors implemented various editorial improvements, refining figure captions, removing redundant text, and ensuring consistency in their descriptions. Given these thorough revisions, I have no further suggestions within the scope of the existing study.

(Remarks on code availability)

Reviewer #2

(Remarks to the Author)

Thank you for responding to the reviewer's comments in this manner. I have no further comments.

(Remarks on code availability)

Reviewer #3

(Remarks to the Author)

Thank you, the authors have satisfied my comments with this revision and appreciate the additional analysis performed which strengthens the manuscript

(Remarks on code availability)

Point-by-Point Response to Reviewer Comments

We thank all reviewers for their insightful questions and their constructive comments. Thank you for the positive feedback on our study for the “novel approach” (R1) and “well thought-out” (R2) study design that “adds significant value to the field of dermatology” (R1). We also appreciate that our paper was found to be “well-executed” (R2) and “well-reported” (R2), and acknowledge that the “robust in person validation” (R3) was valued as strength of our study.

Reviewer #1

Reviewer comment: While the study discusses increased fixation counts in cases of disagreement between dermatologists and AI, further elaboration on how cognitive load specifically interacts with trust in XAI systems would strengthen the interpretation. More detail on how the XAI explanations alleviate or redirect cognitive load would be valuable.

Thank you for your comment. We appreciate your suggestion regarding the interaction between cognitive load and trust in XAI systems. However, we did not measure trust as an endpoint in this study, so we are unable to directly comment on this relationship based on our data. Additionally, due to the study design, isolating the specific effect of XAI explanations on cognitive load was not feasible. Instead, our findings focus on the observed increase in fixation counts in cases of disagreement, and in diagnostically challenging cases, which may suggest increased cognitive effort. We acknowledge the broader relevance of cognitive load and trust in XAI research and have highlighted this as an area for future investigation (red text in the last paragraph of the Discussion).

Reviewer comment: The dataset primarily consists of biopsy-verified images, which ensures high-quality input. However, the study could benefit from discussing how these results generalize to broader clinical settings where input data might not always be as clean. Consider including limitations regarding this in the discussion.

We acknowledge the importance of discussing generalizability. Since biopsy-verified cases are inherently challenging (as they were selected for biopsy/excision due to diagnostic uncertainty), our results are particularly relevant to difficult cases. However, they may not fully generalize to cases that are clearly benign. We have added a statement about this in the limitations section (blue text).

Reviewer comment: While there is a brief mention of experience levels, a more detailed exploration of how dermatologist experience influenced their interaction with XAI

(beyond fixations) would enrich the findings. Breaking down diagnostic accuracy improvements across experience brackets could provide more actionable insights for clinical implementation.

Thank you for the suggestion. We analyzed the impact of dermatologists' experience on diagnostic accuracy improvements when using XAI but found no significant difference between the different experience groups (Fig 2b).

Reviewer comment: The study employs both webcam-based and dedicated eye-tracking methods, which is commendable for cross-validation. However, webcam-based eye tracking is inherently less precise due to variations in participants' environments (e.g., lighting, seating position) and the lower resolution of consumer webcams. These factors can lead to less accurate fixation data, which could introduce noise or error in the interpretation of results. It may bear discussing this as a limitation. Additionally, how much discrepancy exists between the webcam and dedicated systems? Reporting metrics such as fixation accuracy, calibration reliability, or the number of discarded data points from the webcam-based system would enhance transparency.

Thanks for bringing up this important point. We agree that webcam-based eye tracking is more sensitive to external parameters like lighting conditions, seating positions, etc. The eye tracking software we used accounts for some of these discrepancies by pausing recording and alerting participants if the head position deviates beyond the required position. Additionally, it estimates the recording accuracy of the session based on the webcam sampling rate and amount of gaze detected on-screen. Participants with low estimated recording accuracy were excluded to ensure data reliability.

To mitigate the impact of lighting conditions, we instructed participants to sit facing a light source and to avoid backlighting (e.g., open windows behind them). We have added details on these measures under "Study Design and Eye Tracking" in the Methods section (blue text) and provided a supplementary table summarizing discarded participants (Supplementary Table 4, blue text).

Reviewer comment: Understandably, the authors interpret longer fixation durations as indicative of cognitive load and diagnostic difficulty. However, fixation duration can also be influenced by other contextual factors, such as distraction, unfamiliarity with the interface, confusion, or interest in specific visual features. The authors do not account for these alternative explanations, but it may bear mentioning the contextual nature of fixations in the limitations.

While our analysis interprets fixation durations and counts primarily as indicators of cognitive load and diagnostic difficulty, we acknowledge that other factors may also contribute to these observations, primarily unfamiliarity with the interface. From our experience in previous studies

and from the in-person device-based study in this work, we know that participants do not spend much time examining the lesion unless it causes diagnostic uncertainty (they are not sure of their diagnosis). Moreover, the eye tracking software we have used accounts for instances of distraction. As mentioned in the previous comment, it requires the participant to keep their head in a fixed position, stopping measurements and displaying an alert when they deviate.

Reviewer comment: The manuscript mentions that expert dermatologists exhibit fewer fixations, which is consistent with the literature on visual search patterns among experts. However, further comparison between novice and expert behavior could be useful pertaining to their interactions with the XAI explanations? Are experts more likely to ignore AI predictions, or do they verify them in a different manner than novices? Perhaps these are best answered in a future study, but they are compelling questions that are not answered in the current manuscript.

Thanks for your suggestion. As you have pointed out, answering these questions is out of scope for this study. Answering these questions in a systematic manner would require additional survey questions where the dermatologists are asked about their reasoning for the diagnostic decisions and ignoring/following AI decisions. Then we could find differences in AI/XAI interaction behavior between experts and novices. In the future, we plan to conduct such a study.

Reviewer #2 (ML, biomedical decision support):

Reviewer comment: First, in the abstract, the authors claim that "[t]hese insights have significant implications for the design of AI tools for visual tasks in dermatology", but do not follow up in any significant manner (pun intended) by pointing out what these implications are (there is only a vague mention of "[t]raining programs for dermatologists might benefit from incorporating visual fixation training"---unclear what this may be---in the discussion). Please improve.

Thank you very much for your suggestions. We agree that further elaboration on the implications was necessary. Our primary finding was that the addition of text and region-based explanations leads to a measurable increase in diagnostic accuracy compared to AI predictions alone. This highlights the benefit of XAI for improving clinical decision making, which is why we mentioned "significant implications." We added additional text in the Discussion section explaining the implications of increased accuracy, what we meant by visual fixation training, and also added some text on AI interface design (magenta text).

Reviewer comment: Mainly in the introduction, and less so throughout the manuscript, there are sentences that seem overly constructed to the point of either being obvious or meaningless. One such example is "A diagnosis assistance system requires local, strongly end-user-focussed explanations as dermatologists need to assess the quality of the machine suggestions on a case-by-case level". It may be nit-picking, but I encourage the authors to put themselves in the readers' shoes: What is a "local explanation"? It is a given that an explanation in a visual discipline such as dermatology will have to point out characteristic features of a lesion, most of which are local. So I assume this means pointing out the presence of local features. What is "end-user-focussed [sic]"? Directed at the user? Who else would it be directed at? "assess the quality [...] on a case-by-case level": How else to assess than at the level of the individual lesions?

Thank you very much for your suggestions. We agree that parts of the Introduction needed more context for readers unfamiliar with the machine learning field. We used the term "local explanation" (explaining one image) because explanations can also be global (explaining the entire training dataset). We used the term "end-user-focused" because certain types of explanations are model developer focused instead of the end-user (physician). We used the term "on a case-by-case level" because dermatologists may also want explanations that explain the entire model, example: a black blotch is indicative of melanoma. Our feedback from dermatologists has been that they want case-by-case explanations instead of global ones, so we made that distinction in the Introduction. In contrast to local case-by-case explanations, this paper is a good example of global explanations: *Ghorbani, A., Wexler, J., Zou, J. Y. & Kim, B. Towards automatic concept-based explanations.*

We have edited the Introduction to make it easier to understand for non-experts (green text).

Reviewer comment: Another example, which is very minor, but should point out the need to go over the manuscript with a fine-toothed comb: "Their work, however, analyzed pathologists and AI independently, rather than in tandem. Consequently, eye tracking technology can precisely capture where and for how long dermatologists focus their visual attention while using XAI systems." How is the second sentence related to the first via "consequently"?

Thanks for the feedback. We have removed the word "consequently."

Reviewer comment: The authors set out to "[gain] insights into the visual patterns and diagnostic strategies employed by dermatologists when utilizing AI", which is a very interesting goal. However, they "only" (not to diminish their contribution) report on number and duration of eye tracking fixations, which is (arguably) an entirely different thing.

Thank you for your feedback. We understand your concern that fixation metrics, in isolation, may not directly capture the full scope of diagnostic strategies. We would like to clarify that the number and duration of fixations are well-established, foundational measures in eye-tracking studies (example review). While they do not alone capture all higher-level cognitive processes, they provide an objective starting point to understand how dermatologists visually inspect images.

We acknowledge that truly understanding "diagnostic strategies" is more complex than eye-tracking alone can capture. Future research might combine these quantitative metrics with more in-depth qualitative methods (e.g., think-aloud protocols, questionnaires) to provide a more comprehensive picture of diagnostic reasoning.

We revised the text in the Discussion to clarify the scope and limitations of using eye-tracking fixation metrics, and outline how our results form an initial framework for understanding visual attention (grey text in the last paragraph of the Discussion).

Reviewer comment: The authors report that "[t]he explanations consisted of the characteristics that are relevant in diagnosing melanoma/nevus including polygon-based region indications of the detected characteristics", but the caption of Fig 1c gives an explanation of pseudopods that is inconsistent with the polygon that marks the central region of the lesion. Please clarify.

Thank you for your comment. We agree that the polygon does not strictly outline the border of the lesion but also highlights other areas, such as the center. The current model version may highlight broader regions rather than focusing solely on the relevant area. We chose to display a random case as the primary figure instead of cherry picking.

Reviewer comment: The authors refer to "two devices: one with a webcam-based eye tracker and another with a dedicated eye tracker". Later on, the term "device" as in "device-based" is reserved for the dedicated eye tracker. This is confusing at first, so please aim for more consistent terminology.

Thanks for the suggestion. To improve clarity and consistency, we have added the word "device" wherever we refer to the dedicated eye tracker (purple text throughout the manuscript). Additionally, we have removed the sentence about "two devices" from the caption of Figure 1 (crossed-out text in Figure 1 caption).

Reviewer comment: This is major: I am not sure whether the statistical analysis by using a t-test is correct, when in essence the comparison is between two proportions. Please refer to the paper "L.D. Nelson and J.P. Simmons, Moniker maladies: When names sabotage success, Psychol. Sci. 18(12) (2007), pp. 1106–1112." and its subsequent debunking in "B.D. McCullough and T.P. McWilliams, Baseball players with the initial "K" do not strike out more often, J. Applied Statistics 37(6) (2010), pp. 881-891" for a thorough analysis of an instance in which the wrong test was used.

Thank you very much for this insightful question. We assume you are referring to the t-test used to compare the fixation counts when disagreeing with AI/XAI. In the L.D. Nelson paper you cited, methodological flaws – highlighted in the B.D. McCullough paper – invalidated their results, particularly due to their incorrect use of a t-test for comparing two proportions. However, our analysis differs fundamentally in that we do not compare proportions.

A two proportions test is appropriate when data points are binary (e.g., success/failure, yes/no, no strike-out/strike-out, as in the L.D. Nelson paper). If instead we were analyzing the proportion of dermatologists who disagreed versus agreed with the AI, then a test of two proportions could be relevant. However, in our case, we are comparing fixation counts, which are continuous numerical values, during agreement and disagreement with AI. Given the nature of our data, the t-test remains a valid statistical choice. All analyses in the paper were conducted in consultation with an experienced biostatistician (THL).

Reviewer comment: Please explain why there are only discrete balanced accuracy levels in Fig. 2a. I assume it is for clarity of presentation, but it should be mentioned nonetheless.

Thanks for the question. Yes, the discrete balanced accuracy levels were chosen for clarity. Including more levels made the figure overly cluttered, reducing readability. We have added a clarification in the caption of Fig 2a (green text).

Reviewer comment: There is one detail of the diagnostic process missing, or I might have overlooked it: Do the physicians first and independently diagnose the lesion, and later get the (X)AI diagnosis, or do they see the (X)AI diagnosis before making their own diagnosis?

Thanks for the question. The physicians saw the lesion image and the AI/XAI diagnosis simultaneously. To clarify this, we have added additional text in the "Study Design and Eye Tracking" section of the Methods (yellow text).

Reviewer comment: This is major: The authors want to validate webcam-based fixation numbers by later also using a dedicated eye-tracking device. This seems like a good idea to me. When the numbers indicate that outputs of the two systems do not align at all (8.2 vs. 19.2 for AI/agreement, 13.4 vs. 24.9 for AI/disagreement, 9.8 vs. 22.6 for XAI/agreement, 17.1 vs. 28.7 for XAI/disagreement), there is little to no mention of this in the discussion, when it seems that this observation alone may invalidate most of the findings in the webcam-based part of the study.

Thank you for your insightful comment. We acknowledge the discrepancy in absolute fixation numbers between the webcam-based and dedicated eye-tracking device. However, our primary focus is on relative trends rather than absolute values. Despite differences in fixation counts, the overall pattern remains consistent across conditions. For example, in both systems, fixation numbers are higher for disagreement compared to agreement (e.g., for AI: 8.2 vs. 19.2 in the webcam study and 13.4 vs. 24.9 in the device study; for XAI: 9.8 vs. 22.6 in the webcam study and 17.1 vs. 28.7 in the device study).

This consistency in relative trends suggests that while absolute values differ due to differences in tracking methodology, the observed comparative effects across conditions remain meaningful. To ensure proper interpretation, we have added a clarification in the Discussion (green text).

Reviewer comment: "Diagnostic difficulty" and "label difficulty" are used interchangeably. Please use the same term of the same concept.

Thanks for the suggestion. We have changed "label difficulty" to "diagnostic difficulty" (Magenta text under "Diagnostic disagreement between different dermatologists correlates with ocular fixations" in "Results").

Reviewer comment: In the methods: "Using biopsy-verified images ensured that the images in the work were sufficiently challenging for diagnostic purposes"---what does having a biopsy-verified diagnosis available or not have to do with the difficulty of diagnosing a lesion"? Please clarify or rephrase.

Thanks for the question. In the HAM10k dataset, some lesions were biopsied because the examining dermatologist either identified them as clear melanomas or found them uncertain and difficult to diagnose. On the other hand, certain lesions were diagnosed through a consensus of multiple dermatologists rather than biopsy, suggesting they were likely easier to classify. Since we only used biopsied images (i.e., histopathological diagnosis available), we inherently excluded many of the easy-to-classify images. However, to avoid potential confusion, we have removed this sentence from the manuscript.

Reviewer comment: There seem to be conflicting numbers as to the distribution of the lesion classes (benign/malignant) in the study: 22% vs. 78% (p. 14), 8 melanomas and 8 nevi (p. 15). Please clarify.

Thanks for the comment. The 22% and 78% numbers refer to the distribution in the entire dataset, and the 8 melanoma and 8 nevi numbers refer to the distribution of the images used in the study. We have restructured the text to make this more clear under “Datasets” in Methods (blue text).

Reviewer comment: "require more annotation time" (p. 15): The rest of the manuscript refers to diagnosis, it is unclear at which stage annotation of lesions take place. Furthermore, I do not see results of this annotation in the manuscript. Please clarify.

Thanks for the comment. We agree that the word “annotation” does not fit here, so we have changed it to “require more time to diagnose” (green text under “Study Design and Eye Tracking” in Methods).

Reviewer comment: The last two paragraphs on p. 16 contain redundant explanations, please tighten the exposition.

Thanks for the comment. We agree that redundancies exist in these paragraphs, so we have removed certain lines from it (crossed-out text under “Study Design and Eye Tracking” in Methods).

Reviewer comment: Again, this is major (but same point as above): The authors state that "[t]herefore, the onsite validation was critical to confirm that the observed eye-tracking patterns were genuinely reflective of the dermatologists' visual attention and decision-making processes, rather than artifacts of the web-based methodology." The numbers seem to indicate that some results really are "artifacts of the web-based methodology". A possible revision of the manuscript needs to focus on getting this point right.

Thank you for your comment. We believe this concern has been addressed in the previous response about the differing numbers between the web-based and device-based study. Additionally, we have removed this line from the Methods section (crossed-out text in the last paragraph under “Study Design and Eye Tracking” in Methods). We have also discussed this point in the Discussion to clarify the relationship between the two tracking systems and emphasize the consistency of observed trends (green text).

Reviewer #3 (AI for dermatology applications):

Reviewer comment: The finding of statistically significant improved diagnostic accuracy with XAI over AI is more challenging to interpret in light of the prior well done study by Chanda et al. (Ref#2 Nat Commun. 2024 Jan 15;15(1):524) that found in a larger study with >100 dermatologists that XAI did not increase diagnostic accuracy over AI alone. The authors provide very reasonable considerations for why this present result may have differed with an improved classifier and different experimental design. However, given the relatively small magnitude improvement in accuracy observed, it does seem like the larger question of whether XAI will be advantageous in a clinically impactful manner for diagnostic accuracy over identically well performing AI alone is not entirely resolved, at least for this specific task of classifying melanomas from nevi.

Thank you for your insightful points. We believe the observed improvement of 2.8 percentage points is meaningful. As shown in our previous study (Ref#2 Nat Commun. 2024), "plain AI support" already offers a decent improvement over "no AI support." Therefore, we anticipated that the additional improvement with "XAI support" compared to "no AI support" would be relatively smaller but still statistically significant.

That said, we acknowledge that the broader question of whether XAI provides a clinically impactful advantage over equally well-performing AI remains open. A definitive answer would require a prospective study conducted in real clinical settings, which we have planned in the future. To clarify this, we have included a statement in the last paragraph of the Discussion section (orange text).

Reviewer Comment: Figure 3a. The figure legend states: "The gray lines between the boxes connect the same dermatologist between the AI and XAI phases, and the black lines connecting the boxes indicate the means across all dermatologists." I do not see any of these lines in this version of the figure in the manuscript

Thanks for noticing this error. These lines were copied over from another boxplot figure but the part about the gray and black lines do not apply here. We have removed them.

Reviewer comment: I appreciate that the authors acknowledge as a potential limitation in the Discussion the plausibility that 'interpretation of the images during the initial phase may have impacted the subsequent interpretation in the second phase.' This was one of my concerns with the study design, as beyond the addition of XAI, there is a theoretical confounding factor of the clinicians essentially getting a second pass at evaluating the

same case within a relatively short timeframe and could apply additive scrutiny to identify some key diagnostic feature initially missed, regardless of the influence of the XAI. I agree that the references 34-37 on the limitations of expert visual memory help to increase confidence in the conclusions and would suggest moving these from Methods to the relevant portion in discussion. At the same time, these references seem to be based on radiology and cytology expert memory studies rather than anything related to dermatologic lesion assessment, and should be more accurately qualified as such as they do not technically provide direct evidence regarding dermatologists' ability to recall skin lesions as is suggested in the manuscript.

Thanks for the comment. We agree that these references do not provide direct evidence, but they provide analogous support. We have moved this text from the Methods section to the relevant part of the Discussion now and revised it to better align with the points you raised (purple text).

Reviewer comment: Related to this, for the on-site device-based validation, was there a similar >2 week interval between the AI and XAI phases as the webcam based groups? I could not tell from the description in the Methods section

Yes the same >2 week interval was maintained for the onsite study. We added a line about this under "Onsite Validation" in Methods (green text).

Reviewer comment: An interesting consideration of explainable AI is whether it engenders trust that could then lead clinicians to accept faulty recommendations that they may have otherwise rejected with an AI without explainability. If possible, would be interested in the subset of cases where the classifier provided a classification discordant with the ground truth ('faulty' recommendation), and whether clinicians did better with AI or XAI support in this subset of cases. Further, would be interesting to see what the effect was on number of fixations for both AI alone and XAI for this subset.

Thanks for your valuable input. Based on your suggestions, we conducted additional analyses on cases where the AI/XAI prediction was faulty. However, we did not find significant differences between the AI and XAI phases. We have added additional text on this in the Results section under "Disagreements with the classifier decisions correlate with ocular fixations" (blue text).

Reviewer Comment: The lack of correlation between dermatologist experience and increase in diagnostic accuracy with XAI over AI was notable as related prior work had previously identified a correlation between dermoscopy experience and increase in accuracy with XAI over AI (Ref#2 Chanda et al.). The present result suggests that there may be some other clinician factor(s) beyond dermoscopic expertise that might predict

which clinicians may derive more benefit in diagnostic accuracy from XAI over AI alone. While it is beyond the scope of this present study design, it would be informative if the authors might be willing to speculate on what other clinician-specific factors beyond dermoscopic experience might be influencing who benefits “more” from XAI in terms of improved diagnostic accuracy (? Training background, baseline attitudes towards AI, etc), and how this might be explored in future work

Thanks for bringing up this important point. Our previous study did indeed find a correlation between dermatologists’ experience and their benefit with XAI over AI. However, there is a minor difference between how experience was defined in this study versus the prior study. In the prior study, we defined experience as their frequency of use and involvement in dermoscopy, i.e. rare use, occasional use, regular use, regular use and involvement in science. In contrast, in this study, we measured experience by their actual years of experience as dermatologists regardless of their frequency of use. This might explain the difference in findings between the two studies. To clarify this, we have added an explanation in the Discussion section (orange text).

Reviewer comment: minor suggestion for clarity: The last paragraph of Results that begins “In summary, our analysis reveals that disagreements between dermatologists and classifier predictions correlate with increased ocular fixations...” is confusing in that it is presented within the “Diagnostic disagreement...” subsection but actually summarizes the entire Results section. This could easily be addressed in copy editing with appropriate spacing, or if not possible consider removal as all of the key results are addressed in the Discussion anyway

Thanks for the suggestion. We agree with this and have removed the final paragraph for clarity (crossed-out text under “Diagnostic disagreement between different dermatologists correlates with ocular fixations” in the Results section).